# The Influence of Envelope C-Terminus Amino Acid Composition on the Ratio of Cell-Free to Cell-Cell Transmission for Bovine Foamy Virus

**DOI:** 10.3390/v11020130

**Published:** 2019-01-31

**Authors:** Suzhen Zhang, Xiaojuan Liu, Zhibin Liang, Tiejun Bing, Wentao Qiao, Juan Tan

**Affiliations:** Key Laboratory of Molecular Microbiology and Technology, Ministry of Education, College of Life Sciences, Nankai University, Tianjin 300071, China; zhangsuzhen819819@163.com (S.Z.); 1395312330@qq.com (X.L.); liangzhibin@mail.nankai.edu.cn (Z.L.); btj1987@126.com (T.B.); wentaoqiao@nankai.edu.cn (W.Q.)

**Keywords:** bovine foamy virus, infectious clone, particle release, cell-free transmission

## Abstract

Foamy viruses (FVs) have extensive cell tropism in vitro, special replication features, and no clinical pathogenicity in naturally or experimentally infected animals, which distinguish them from orthoretroviruses. Among FVs, bovine foamy virus (BFV) has undetectable or extremely low levels of cell-free transmission in the supernatants of infected cells and mainly spreads by cell-to-cell transmission, which deters its use as a gene transfer vector. Here, using an in vitro virus evolution system, we successfully isolated high-titer cell-free BFV strains from the original cell-to-cell transmissible BFV3026 strain and further constructed an infectious cell-free BFV clone called pBS-BFV-Z1. Following sequence alignment with a cell-associated clone pBS-BFV-B, we identified a number of changes in the genome of pBS-BFV-Z1. Extensive mutagenesis analysis revealed that the C-terminus of envelope protein, especially the K898 residue, controls BFV cell-free transmission by enhancing cell-free virus entry but not the virus release capacity. Taken together, our data show the genetic determinants that regulate cell-to-cell and cell-free transmission of BFV.

## 1. Introduction

Foamy viruses (FVs), also known as spumaviruses, are a group of *Retroviridae* with unique features that differentiate them from orthoretroviruses. FVs infect humans [1,2] and other mammals, including bovines [3], simians [4], felines [5], and equines [6]. However, FV infection does not cause any clinical symptoms in its natural hosts, despite the significant cytopathic effect it causes in fibroblasts or fibroblast-derived cell lines as well as in epithelial cells, such as baby hamster kidney (BHK) cells [7,8].

Viruses have two major transmission strategies: cell-free transmission, involving the release of virus particles into the extracellular space, and cell-to-cell transmission [9,10,11]. Retroviruses exhibit different degrees of cell-free and cell-to-cell transmission. Unlike most other retroviruses, such as the human immunodeficiency virus (HIV) [12,13,14,15,16], murine leukemia virus (MLV), feline foamy virus (FFV), prototype foamy virus (PFV), and simian foamy virus (SFV), which transmit through both cell-to-cell and cell-free pathways, bovine foamy virus (BFV) infection is tightly cell-associated [17,18].

In contrast to other retroviruses, the envelope (Env) protein of PFV plays an important function in the budding and release of PFV particles [19]. In particular, the leader peptide (LP) in the N-terminal region of PFV Env is essential for virus budding. In LP, the three lysine residues (K14, K15, and K18) undergo ubiquitination, which regulates PFV release [20].

The Env protein determines FV’s wide host range [1,2,3,4,5,6]. The cellular receptor of FVs has not been determined; however, it was reported that heparin sulfate might act as an attachment factor facilitating PFV and SFV entry [21,22]. Different from orthoretroviruses, the assembly and budding of FV particles require direct and specific interaction between the N-terminus of Gag and the Env leader protein Elp [23,24]. FV Gag, lacking the myristoylation membrane targeting signal, cannot produce cell-free Gag-only virus-like particles [18,24,25]. Instead, co-expression of FV Gag and Env leads to the generation of Env-dependent sub-viral particles (SVPs), which sets FVs apart from orthoretroviruses [23,24,26,27,28].

Bao and colleagues selected high-titer (HT) cell-free BFV-Riems isolates using the in vitro evolution procedure [18]. Yet, they did not generate infectious viral DNA clones and did not explore the molecular mechanisms that have enabled BFV cell-free transmission. Using the BFV strain BFV3026, which we isolated in 1996, we generated an infectious clone called pBS-BFV-B [29]. BFV-B is deficient in cell-free transmission, which does not allow for the development of a BFV vector. We have now screened for BFV variants with enhanced cell-free transmission in BICL cells (derived from BHK-21 cells) by serial virus passaging and successfully created a BFV infectious clone—called pBS-BFV-Z1—with cell-free transmission ability. Through sequence alignment and mutagenesis, we determined the C-terminal region of Env as one determinant for BFV cell-free transmission, and thus uncovered the molecular mechanism by which BFV spreads via cell-free transmission.

## 2. Materials and Methods

### 2.1. Cell Lines and Viruses

BHK-21, Cf2Th, HEK293T, BFVL (BHK21-derived indicator cells containing a *luciferase* gene under the control of the BFV LTR) [30], and BICL (BHK21-derived indicator cells containing an enhanced green fluorescent protein under the control of the BFV LTR) cells [31,32] were maintained in Dulbecco’s modified Eagle’s medium (Thermo Fisher, Waltham, MA, USA) containing 10% fetal bovine serum (GE Healthcare, Cincinnati, OH, USA) and 1% penicillin-streptomycin (Thermo Fisher, Waltham, MA, USA) at 37 °C in a 5% CO_2_ atmosphere. BFV3026 was stored in our lab and cultured with Cf2Th and BICL cells. No mycoplasma and viruses contamination were detected in any cells we used.

### 2.2. Plasmids and Transfection

BFV3026 full-length genomic DNA clone pBS-BFV-B was generated by amplifying viral DNA extracted from BFV3026-infected Cf2Th cells. The BFV infectious clone pBS-BFV-Z1 was constructed using the same methods of pBS-BFV-B as previously reported [29]. Chimeric BFV clones between clone B and Z1 were generated by shared different restriction sites. Mutations were generated using site-direct PCR mutagenesis (Toyobo, Osaka, Japan), and all mutations were verified by DNA sequencing (Genewiz, Beijing, China). The plasmids expressing Env and Gag were constructed by inserting the coding sequences of BFV Env and Gag into the indicated vectors, including pCMV-3HA and pCE-puro-3Flag. HEK293T and BHK-21 cells were transfected using polyethylenimine (PEI) (Polysciences, Warrington, PA, USA) according to the manufacturer’s protocol [33]. BHK-21 cells (2 × 10^5^) were seeded in six-well plates. Twenty-four hours later, 2 μg empty vector pBS, pBS-BFV-B, pBS-BFV-Z1, or chimeric infectious clones were transfected into BHK-21 cells. Eight μL PEI (1 mg/mL) was added at DNA:PEI (μg:μg) ratios of 1:4 and incubated for 10 min at room temperature. Cells were harvested 48 h after transfection. Cf2Th cells were transfected using Lipofectamine 2000 (Thermo Fisher, Waltham, MA, USA).

### 2.3. Titration of Cell-Free BFV3026

BFV3026 was isolated from lymphocytes of peripheral blood of cattle by our lab in 1996 and cultured with Cf2Th cells. The parental cell-free BFV3026 virus was obtained by freezing-thawing highly infected Cf2Th cells. The later serial passage screening was carried in BICL cell lines. BFV was adapted to cell-free transmission in BICL cells by serial passaging using cell-free culture supernatants as previously reported [18]. In the case of viral infection, BICL cells expressed GFP and green signals were visible under a fluorescent microscope.

BFV cell-free titers were determined by infecting BICL indicator cells with gradient diluted BFV-containing supernatants. BICL cells were seeded (2 × 10^4^ cells/well) in 48-well plates. Twenty-four hours later, BFV-containing supernatants were serially diluted 1:5 (triplicate for each dilution) and added to BICL cells. Six days post-infection, BFV titers were determined by fluorescence microscopy for GFP signals in BICL cells. If at least two of the three wells were positive for GFP, that dilution was considered positive for infection.

### 2.4. Cell-Free Infection

BHK-21 cells were transfected with indicated infectious clones for 2, 4, and 6 days. Then, cell culture supernatants were collected and filtered with a 0.45 μm membrane. BFVL cells in 12-well plates (2.5 × 10^5^ cells/well) were then infected with equal volumes of these filtered supernatants for 48 h and subjected to luciferase assay. The infectivity of the cell-free virus was determined by luciferase activity.

### 2.5. Cell Co-Culture Assay

BHK-21 cells were transfected with indicated infectious clones for 2, 4, and 6 days. Then, the 5% of cells were harvested and co-cultured with 1.5 × 10^5^ BFVL cells in 12-well plates. Forty-eight hours post co-culture, luciferase activity was measured to quantitate virus titer.

### 2.6. Luciferase Reporter Assay

Forty-eight hours after infection or co-culture, the BFVL cells were harvested in lysis buffer. Then, luciferase assays were performed using the luciferase reporter assay system (Promega, Madison, WI, USA). In addition to normalizing the luciferase activities, we also detected the GAPDH expression of cell lysates by western blotting. All of the results from the luciferase reporter assay were the averages of three independent experiments.

### 2.7. Hirt DNA Extraction

BICL cells infected with cell-free BFV p43 (passage 43) were harvested, washed with 1.5 mL 1× PBS, and lysed with 250 μL buffer K (20 mM HEPES, 140 mM KCl, 5 mM MgCl_2_, 1 mM Dithiothreitol) and 7.5 μL 0.5% Triton X-100 for 10 min at room temperature. Following centrifugation (10 min at 1500× *g*), the pellet was dissolved in 400 μL TE and 90 μL 5 M NaCl at 4 °C overnight. The supernatant post centrifugation was extracted with phenol chloroform: isoamylalcohol (24:1), then precipitated for DNA in ethanol with 0.3 M NaAc at −20 °C for 1 h. The purified DNA was then washed with 70% ethanol, dissolved in 20 μL TE, and stored at −20 °C.

### 2.8. Enrichment of Wt and Sub-Viral BFV Particles

Six milliliters of cell culture supernatants containing BFV particles or SVPs (including Env-only and Gag-Env SVPs) were filtered through 0.45 μm membranes and layered on a 1 mL cushion of 20% sucrose in PBS (*w*/*v*). After ultracentrifugation (Optima LE-80K, Beckman Coulter) at 4 °C for 1.5 h at 210,053× *g*, the non-visible pellet was re-suspended in 30 μL loading buffer containing 2% SDS and subjected to western blotting.

### 2.9. Co-Immunoprecipitation

HEK293T cells transfected with pCMV-3HA-Gag and pCMV-3HA-Env were lysed in lysis buffer (50 mM Tris, pH 7.4; 150 mM NaCl; 2 mM EDTA; 3% Glycerol; 1% NP-40; complete, EDTA-free protease inhibitor cocktail tablets). The cell lysates were sonicated and centrifuged at 13,000× *g* for 10 min at 4 °C. Following centrifugation, supernatants were incubated with mouse anti-Gag antibody for 2 h at 4 °C and rotated with Protein A-agarose (Merck Millipore, Darmstadt, Germany) for 3 h or overnight at 4 °C. After six washes with lysis buffer, the immunoprecipitated materials were boiled in 40 μL of 2× SDS loading buffer and subjected to western blotting using rabbit anti-HA antibody.

### 2.10. Immunofluorescent Assay

BHK-21 cells seeded in 12-well plates containing coverslips were infected with infectious clones. After 48 h, BHK-21 cells were fixed with 500 μL of 4% formaldehyde in PBS for 10 min on a shaker. Cell membranes were permeabilized with 500 μL of penetration solution (0.1% Triton X-100 in 4% formaldehyde) for 10 min. Then, cells were blocked in 500 μL of blocking solution (50% serum, 6% non-fat milk power, 15% BSA, 5% NaN_3_, 20% Triton in PBS) for 2 h at room temperature or 4 °C overnight. Cells were then incubated in diluted primary antibody for 2 h at room temperature. The coverslips were washed in the 12-well plates with PBST and incubated in diluted fluorochrome-conjugated secondary antibody for 40 min in the dark. After 5 washes, cells were incubated in 500 μL of DAPI for 10 min in the dark. The coverslips were mounted onto glass slides and allowed to air dry for 1 h. For long-term storage, slides were stored flat at 4 °C protected from light.

### 2.11. Western Blotting

Proteins were separated by SDS-PAGE (polyacrylamide gel electrophoresis). Then, the separated proteins were transferred onto a polyvinylidene difluoride (PVDF) membrane (GE Healthcare, Cincinnati, OH, USA) by electroblotting for 1 h at 100 V in 4 °C. Following incubation in PBS containing 5% nonfat milk for 45 min at room temperature, the PVDF membranes were subsequently incubated with primary antibody for 90 min. Then, the membranes were incubated with secondary antibodies—goat anti-mouse or goat anti-rabbit IgG conjugated to horseradish peroxidase, for an additional 45 min. Protein bands were detected by chemiluminescence (Merck Millipore, Darmstadt, Germany).

### 2.12. Statistical Analysis

Data were expressed as the mean ± SD of the results of three independent experiments in which each assay was performed in triplicate. Data were compared using the unpaired two-tailed *t* test. A *p* value of <0.05 was considered significant (* *p* < 0.05, ** *p* < 0.01, *** *p* < 0.001, **** *p* < 0.0001).

## 3. Results

### 3.1. Construction of a Cell-Free BFV Infectious Clone pBS-BFV-Z1

To isolate a high-titer cell-free BFV stain, we performed in vitro evolution experiments in BICL cells, which express GFP upon BFV infection. The original virus strain used for selection was Cf2Th-associated BFV3026, which was isolated in China. After 53 passages, the cell-free infectivity of BFV reached a plateau of 10^5^ IU/mL (Figure 1A). The full-length viral genomic DNA was amplified from the hirt DNA extracted from BICL cells infected with cell-free BFV (passage 43) and then inserted into the pBluescript SK-(pBS) vector. The constructed infectious viral DNA clone was named pBS-BFV-Z1.

We next characterized the pBS-BFV-Z1 clone for its ability to produce infectious cell-free particles by harvesting viruses in the supernatants of BHK-21 cells that were transfected with BFV infectious clones. As shown in Figure 1B, the BFV particle release was observed in Z1-transfected cells but not in cells transfected with the cell-associated clone, pBS-BFV-B. At the same time, we also analyzed the intracellular Gag expression levels and observed that Gag expression in BFV-B was less than BFV-Z1 (Figure 1B). Furthermore, we observed that the replication capacity of the Z1 clone was 40 times greater than that of the B clone, as measured by co-culture assay (Figure 1C). Notably, the cell-free Z1 virus particles were infectious (Figure 1C) and spread in a long-term infection (Figure 1D). Together, these data demonstrate that the pBS-BFV-Z1 clone is highly infectious and produces infectious cell-free virus particles.

### 3.2. The C Terminus of Env Determines the Ability of BFV to Generate Infectious Cell-Free Particles

To identify which regions in the sequence of pBS-BFV-Z1 changed and enabled the production of infectious cell-free virus particles, we aligned the sequences of the two BFV clones, B and Z1. Changes were identified at multiple nucleotide positions (Appendix A). We then swapped the sequences between the B and Z1 clones and generated six chimeric BFV clones (*Eco*R I within *pol* gene, and *Nde* I within *env*) (Figure 2A). As shown in Figure 2, clone S1S2 and clone Z1 produced similar levels of intracellular Gag (Figure 2B) as well as comparable cell-to-cell transmission activity (Figure 2D). Unfortunately, none of these chimeric clones produced infectious cell-free BFV particles (Figure 2C). Notably, the expression of Gag in BFV-B could be detected for two days (Figure 1B) but not for four days that was passaged one time with a low ratio of transfected cells (Figure 2B). One possibility is that the activity of clone B is much weaker than clone Z1 (Figure 1C).

Previous studies have shown that Env protein is essential for PFV release and entry [26,34,35]. Clone S1S2 contains a chimeric *env* gene, including the N terminus of EnvZ1 and the C terminus of EnvB, but lacks cell-free transmission ability. To test the role of Env in the ability of the Z1 clone to produce infectious cell-free virus particles, we engineered another two chimeric clones, Z1(1–9554) and Z1(EnvcB). Clone Z1(1–9554) contained an entire *env* gene from clone Z1, whereas clone Z1(EnvcB) had the clone Z1 sequence except for the C-terminal region of Env, which was derived from clone B (Figure 3A). The intracellular Gag expression from Z1(1–9554) and Z1(EnvcB) was similar to that from clones Z1 and S1S2 (Figure 3B). Excitingly, the Z1(1-9554) clone produced high levels of infectious cell-free virus particles, although still two to three-fold lower than the parental clone Z (Figure 3C). In contrast, the cell-to-cell transmission of Z1(EnvcB) was two-fold more efficient than that of clone Z1, as measured by co-culture assay (Figure 3D). However, compared with clone Z1, chimera Z1(EnvcB) was unable to transmit by cell-free particles, which suggests that replacement of the C-terminal Env with that from clone B abrogated the cell-free infectivity of clone Z1 (Figure 3C). Furthermore, Z1(1–9554) exhibited cell-free transmission activity, although to a lesser degree than Z1 (Figure 3C). Overall, these results indicate that the C-terminus of Env is crucial for BFV cell-free infectivity.

To identify the specific sites that affect the cell-free transmission ability of clone Z1, we compared the amino acid sequence of the C-terminal region of Env between clone Z1 and clone B and found differences at five sites: H816Y, P823S, E898K, R976K, and L978S (Figure 4A), which are all located in the C-terminus of transmembrane (TM) protein gp48 (aa 572–990). Using chimera S1S2 that bears the C-terminal region of the clone B Env, we mutated the amino acids at the above five positions to the counterparts in clone Z1. Our results showed that two single point mutations—H816Y and E898K, especially E898K—rendered clone S1S2 to produce infectious cell-free virus particles. Furthermore, the cell-free infectivity of clone S1S2-898/976/978, which had the amino acids at position 898, 976, and 978 of Z1 Env, was similar to that of Z1(1–9554) (Figure 4B). We also mutated the above five positions in clone Z1 Env to the respective amino residues in clone B Env. The K898E mutation markedly impaired the cell-free infectivity of Z1, and mutations Y816H and K898E together abrogated the cell-free infectivity of Z1 (Figure 4D). These results indicate that amino acid K898 together with Y816 in the Env protein are major determinants of BFV cell-free transmission.

### 3.3. The 14-AA Deletion in Gag Gene Increases BFV Cell-Free Transmission Activity

Gag plays an indispensable role in the assembly and release of PFV particles. We found that Gag of the Z1 clone is 14 aa shorter than Gag of the B clone (Figure 5A). We tested whether this 14-aa deletion had an effect on BFV cell-free transmission by constructing two chimeric clones, Z1(Gag60) and Z1(1-9554; Gag60), which had the 14 aa inserted back in clones Z1 and Z1(1-9554). BICL cells were transfected with the above BFV clones and observed similar levels of Gag expression from all viral DNA clones (Figure 5B). However, the cell-free infectivity of Z1(Gag60) was four- to five-fold lower than that of Z1 (Figure 5C), although the cell-to-cell transmission activities of these two clones were comparable (Figure 5D). We further observed that Gag-B, Gag-Z1, and Gag60 were similarly distributed in the cytoplasm, suggesting that this 14-aa sequence does not affect the cellular localization of Gag (Figure 5E).

Unlike orthoretroviruses, the release of FV virions requires both Gag and Env proteins [36]. FV can form Gag-Env SVPs or Env-only SVPs [26,34,37]. We therefore examined the ability of Env-B and Env-Z1 to mediate SVPs release. Different *env* and *gag* genes were cloned into the pCMV-3HA or the pCE-puro-3Flag vector. Gag plasmids alone or together with different Env plasmids were transfected into HEK293T cells. Two days post-transfection, virus particles in the culture supernatants were harvested through ultracentrifugation and analyzed by western blotting. As shown in Figure 6A, SVPs were detected when Gag was expressed together with Env but not when Gag was expressed alone. Similar levels of SVPs were produced by Env and Gag from both clone B and clone Z1. However, Env-Z1, but not Env-B, was completely processed in virus particles.

Next, co-immunoprecipitation (Co-IP) assays were performed to measure the interaction between Env and Gag. 3Flag-Env and 3HA-Gag were co-transfected into HEK293T cells followed by IP with anti-HA antibody. The results of western blots showed similar levels of interaction between Env and Gag of both the B and Z1 clones (Figure 6B). Taken together, the 14-amino acid sequence that is missing in the Z1 Gag contributes to the increased cell-free transmission of clone Z1, but this activity is not a result of greater ability of assembling SVPs nor interacting with Env.

### 3.4. The C-Terminal Region of Env Modulates the Entry of Cell-Free BFV Particles But Does Not Affect Virus Release

We showed that chimeric clone Z1(EnvcB), which contains the C-terminus of Env-B, is defective in cell-free transmission (Figure 3C). To further understand the underlying causes, we examined the ability of EnvcB in mediating the assembly and release of Gag-Env particles. The results showed similar levels of SVPs that were produced with either EnvcB, Env-Z1 or Env-B together with Gag (Figure 7A). In addition, all three Env proteins formed Env-only SVPs (Figure 7A), which is consistent with the previous reports [26,34,37]. We also observed that there were more uncleaved gp130 in Env-B-containing SVPs that were produced with Gag-Env and Env alone compared to Env-Z1 SVPs and EnvcB SVPs, which suggests that this C-terminal region of Env affects Env protein processing.

To determine whether the defective processing of BFV Env diminishes Env entry function, we used BFV Env proteins to pseudotype HIV-1(env-) viruses and measured virus infectivity. BFV Env proteins were co-expressed with HIV-1 DNA clone NL4-3.Luc.env-, which harbors a luciferase reporter gene and does not express HIV-1 Env. The culture supernatants containing BFV Env-pseudotyped HIV-1 particles were used to infect Cf2Th cells. Luciferase expression in Cf2Th cells reflects virus entry mediated by BFV Env proteins. The results showed that both Env-B and EnvcB were able to mediate HIV-1 entry, but the efficiency was 85-fold lower than Env-Z1. Furthermore, the K898E mutation reduced the entry efficiency of Env-Z1, which is in agreement with the increased entry of EnvcB harboring the E898K mutation (Figure 7B). We also tested the efficiency of these Env proteins in mediating cell-cell fusion. HEK293T cells were co-transfected with BTas DNA and different Env DNA and then co-cultured with the BFVL indicator cells. Env-mediated fusion of 293T and BFVL cells allows BTas to activate the expression luciferase reporter gene in BFVL cells. These results showed that EnvcB and EnvB led to much higher cell fusion than EnvZ1 (Figure 7C). The E898K mutation impaired the ability of EnvcB to mediate cell fusion (Figure 7C). Together, these data suggest that the C-terminal region of Env affects Env processing and therefore has a key role in the function of Env to mediate virus-cell or cell-cell fusion.

## 4. Discussion

In this study, we generated high-titer cell-free BFV variants by serial virus passaging in vitro and further constructed a cell-free infectious clone, pBS-BFV-Z1. We also identified the viral genes and key amino acids that regulate BFV cell-free transmission. In particular, we found that the C-terminal region of Env, especially the K898 residue, is essential for BFV cell-free transmission activity.

It is known that BFV is highly cell-associated and spreads mainly through cell-to-cell transmission [17,18]. Although FFV, SFV, and PFV can transmit by both cell-to-cell and cell-free pathways, the sequence similarity between BFV and other FVs (FFV, SFV, and PFV) is low, and it is thus difficult to identify the viral genetic determinants through sequence alignment. We thus performed in vitro evolution to generate high-titer cell-free BFV variants. There are two factors that influence cell-free transmission, the number and the infectivity of the released virions [9,10]. Using the two distinct isolates, BFV-B and BFV-Z1, we detected Env-dependent production of BFV Gag-Env SVPs and demonstrated that Env-B was as efficient as Env-Z1 and EnvcB in mediating SVPs release (Figure 7A). However, Env-B was not completely processed in virus particles when compared with Env-Z1. It is reported that all known FV Env proteins contain a conserved optimal cleavage site RX(K/R)R between the SU and TM subunits and processing of SU/TM does not affect the egress of viral particles but is essential for the infectivity of the released virus particles [26,38,39]. Indeed, Env-Z1 was 85-fold more efficient than EnvcB and EnvB in mediating cell-free virus entry (Figure 7B). Our study demonstrates that the C-terminal sequence of Env regulates BFV cell-free entry efficiency rather than virion release capacity.

BFV Env is essential for virus budding, release, entry, and membrane fusion [18,19]. Compared with BFV-B, there are a few mutations in BFV-Z1 Env (eight changes among 990 residues), including three mutations in the N-terminus and five in the C-terminus. Previous reports state that the ubiquitination of three lysine sites in the N-terminal cytoplasmic tail region of PFV Env protein increases virus infectivity and decreases the production of SVPs [20]. In our study, the four N-terminal mutations (D42N, V55I, and P134H) did not change cell-free virus infectivity (Figure 3C). In contrast, the C-terminal mutations H816Y, P823S, E898K, R976K, and L978S, especially the E898K mutation, enhance cell-free transmission (Figure 4B,D). It is well-known that lysine (K) can undergo methylation, acetylation, succinylation, ubiquitination, and other modifications, which play an important role in regulating the protein activity and structure adjustment. Interestingly, we found that the equivalent position of the 898 residue in four currently described BFV isolates from the United States (GenBank accession number NC001831.1) [40], China (accession number AY134750.1) [41], Poland (accession number JX307861) [42,43], and Germany (accession number JX307862) [43], which spread only through cell-to-cell, is not a lysine (K). Nevertheless, the equivalent position is occupied by a lysine in other high titer cell-free strains, such as SFV and PFV. These suggest that the K898 in Env has an important role in FV cell-free transmission. The underlying mechanism warrants further study.

The Gag protein of BFV-Z1 lost a 14-amino acid sequence compared to BFV-B. This 14-residue deletion led to a smaller Gag-Z1, as shown by the results of western blotting (Figure 5B). Interestingly, Gag-Z1 enhanced cell-free infectivity by four- to five-fold (Figure 5C). Similar deletions have also been reported in other high titer cell-free BFV strains, although there is differing in the number of missing amino acids and their locations (Professor Martin Löchelt’s unpublished data). It is known that the Gag-Env interaction is very important for the budding and release of BFV virions. Yet, in our study, the interaction of Gag and Env in BFV-B and BFV-Z1 was almost the same, which suggests that the contribution of Gag-Z1 to enhanced cell-free transmission is not through promoting interaction with Env. The other changes of the BFV-Z1 genome contributed little to BFV cell-free transmission.

In summary, we demonstrated that the C-terminus of Env, especially the K898 residue, is critical for BFV cell-free transmission. This function of Env C-terminal sequence results from promoting BFV cell-free entry efficiency rather than viral particle release capacity. Our study thus suggests the possibility of generating high-titer BFV vectors through engineering viral Env—in particular, the C-terminal sequence.

## Figures and Tables

**Figure 1 viruses-11-00130-f001:**
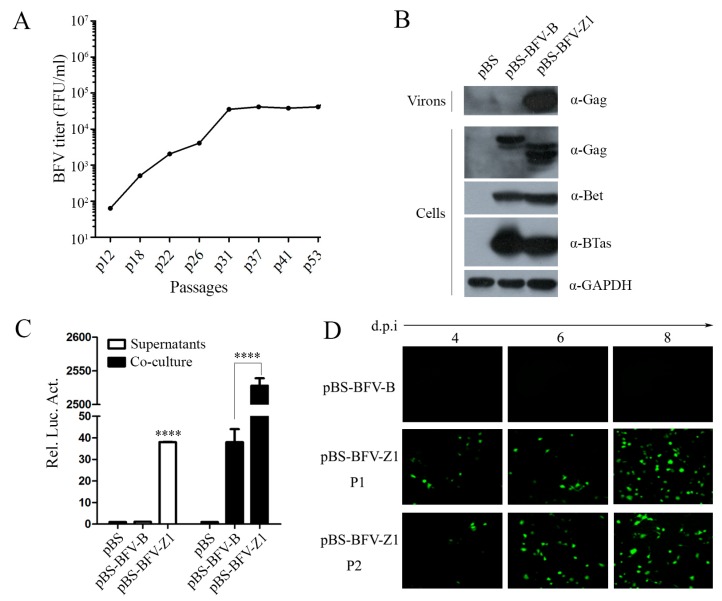
The cell-free and cell-to-cell transmission activity of BFV-Z1. (**A**) Screen of the high-titer cell-free bovine foamy virus (BFV) strain in BICL cells with cleared supernatants. Around 53 passages, BFV cell-free titers were determined by gradient dilution. BICL cells were seeded in 48-well plates (2 × 10^4^ cells/well), then BFV-containing supernatants were serially diluted 1:5 in the plate (triplication for each dilution). Four to six days post-infection, the cell-free BFV titers were calculate and analyzed, as determined by fluorescence microscopy for GFP signals in BICL cells. (**B**–**C**) BHK-21 (baby hamster kidney) cells (2 × 10^5^) were seeded in 6-well plates. After 24 h, 2 μg empty vector pBS, pBS-BFV-B, or pBS-BFV-Z1 were transfected into BHK-21 cells. Four days later, 5% cells co-cultured with 1.5 × 10^5^ BFVL cells and part of supernatants infected 2.5 × 10^5^ BFVL cells, respectively, for cell-to-cell and cell-free transmission activity using a luciferase activity assay after 48 h, and the results were the averages of three independent experiments and data were analyzed using GraphPad Prism software (compared with BFV-B, * *p* < 0.05, ** *p* < 0.01, *** *p* < 0.001, **** *p* < 0.0001). The remainder of the supernatants and cells transfected for 2 days were harvested for western blot with the indicated antibodies (B). (**D**) Part of the supernatants were collected to infect fresh BICL cells (marked P1) prior to ultracentrifugation, and the expression of GFP was observed. If more than 80% BICL cells were positive for green fluorescence, then the supernatants were collected and cleared to infect fresh BICL cells (marked P2).

**Figure 2 viruses-11-00130-f002:**
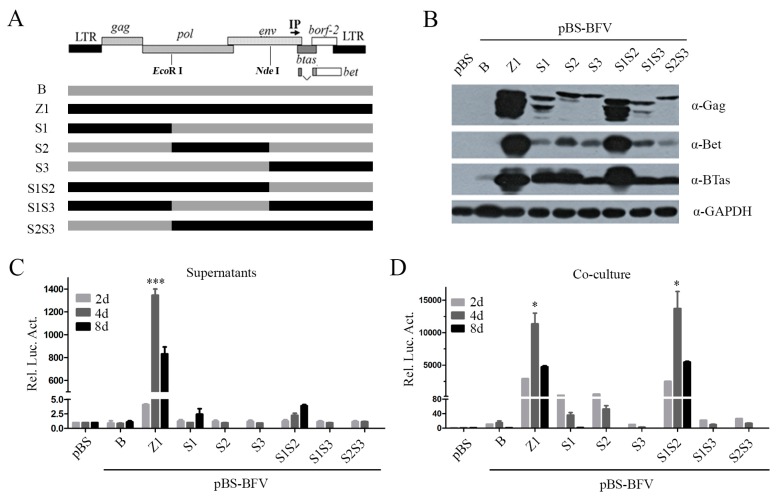
Activities of chimeric BFV infectious clones between pBS-BFV-B and pBS-BFV-Z1. (**A**) The structure model of BFV proviral genome is shown on the top, followed with a schematic overview of the chimeric virus between BFV-B and BFV-Z1. The gray box represents the BFV3026 proviral genomic DNA of pBS-BFV-B, and the black box represents pBS-BFV-Z1. The full-length viral DNA was divided into three segments based on the single restriction enzyme sites of *Eco*R I and *Nde* I, and the new chimeric clones were generated by exchanging the fragments between pBS-BFV-B and pBS-BFV-Z1 plasmids. (**B**–**D**) BHK-21 (2 × 10^5^) cells were transfected with 2 μg empty vector pBS, pBS-BFV-B, pBS-BFV-Z1, or chimeric infectious clones as indicated for 2, 4, or 8 days. Cells transfected for 4 days were subjected to western blotting with indicated antibodies (**B**). The activity of cell-free and cell-to-cell transmission was measured by a luciferase activity assay at the indicated times (**C**,**D**), and the data were the averages of three independent experiments. Compared with BFV-B, * *p* < 0.05, ** *p* < 0.01, *** *p* < 0.001, **** *p* < 0.0001.

**Figure 3 viruses-11-00130-f003:**
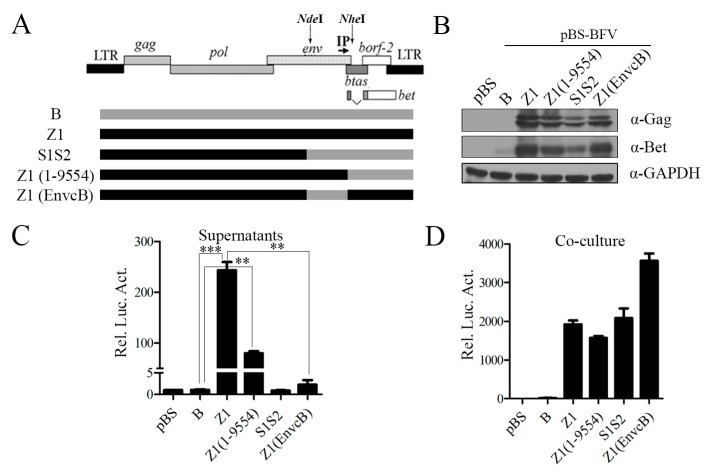
The C terminus of Env determines the ability of BFV to generate infectious cell-free particles. (**A**) We constructed clone BFV-Z1(1-9554) by the single restriction enzyme sites of *Nhe* I, which cuts downstream of Env in the BFV proviral genome, and clone BFV-Z1(EnvcB) was constructed by *Nde* I and *Nhe* I. The gray box represents the BFV3026 proviral genomic DNA of pBS-BFV-B, and the black box represents pBS-BFV-Z1. (**B**–**D**) BHK-21 cells (2 × 10^5^) were transfected with 2 μg empty vector pBS, pBS-BFV-B, pBS-BFV-Z1, pBS-BFV-Z1(1-9554), pBS-BFV-S1S2, or pBS-BFV-Z1(EnvcB) for 4 days, and cells were subjected to western blotting with indicated antibodies (**B**). The cell-free and cell-to-cell transmission activity was measured with the indicated assays (**C**,**D**), and the data were the averages of three independent experiments.

**Figure 4 viruses-11-00130-f004:**
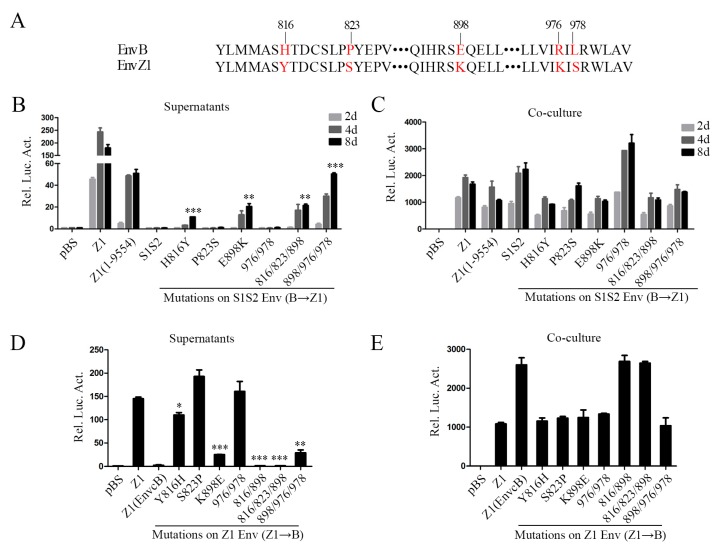
K898 in the *env* gene of BFV-Z1 is crucial for BFV cell-free transmission. (**A**) Amino acid site mutations, marked in red, in the C terminus of Env by sequence alignment analyses are shown. (**B**,**C**) Construction of infectious clones with point mutations on the basis of pBS-BFV-S1S2 from B to Z1 and their transfection to BHK-21 cells. The transmission activity was analyzed by measuring luciferase activity, and the data were the averages of three independent experiments. Compared with BFV-S1S2, * *p* < 0.05, ** *p* < 0.01, *** *p* < 0.001, **** *p* < 0.0001. (**D**,**E**) Construction of clones with point mutations based on pBS-BFV-Z1 from Z1 to B. Then, these were transfected into BHK-21 cells and the transmission activity was analyzed. Compared with BFV-Z1, * *p* < 0.05, ** *p* < 0.01, *** *p* < 0.001, **** *p* < 0.0001. “ㆍ” stands for omission.

**Figure 5 viruses-11-00130-f005:**
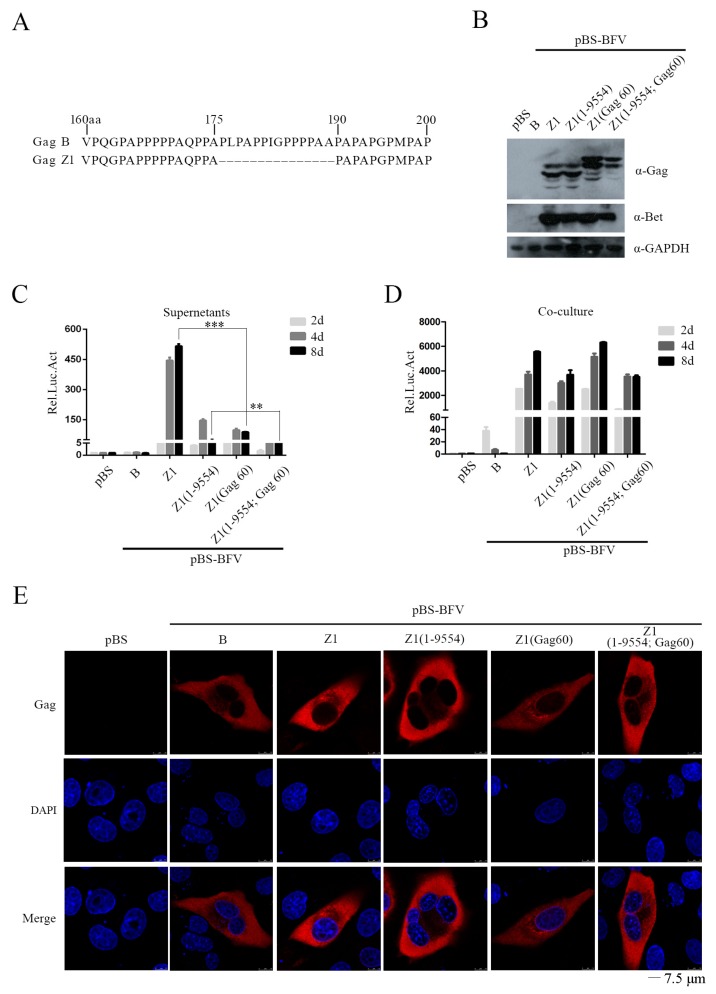
The 14 aa deletion in *gag* gene increases BFV cell-free transmission activity. (**A**) Partial results—sequence alignment of amino acids in Gag-B and Gag-Z1 is shown in the top left corner. Starting with the 176th amino acid, 14 consecutive amino acids are missing in Gag-Z1. (**B**) BHK-21 cells were transfected with empty vector pBS, pBS-BFV-B, pBS-BFV-Z1, pBS-BFV-Z1(1-9554), pBS-BFV-Z1(Gag60), or pBS-BFV-Z1(1-9554; Gag60), and partial lysates of cells after 4 d of transfection were tested by western blotting analysis. (**C**,**D**) The cell-free and cell-to-cell transmission activity was detected separately at the indicated transfection time, and the data were the averages of three independent experiments. (**E**) BHK-21 cells were transfected with indicated infectious clones, then subjected to immunofluorescent assays. Confocal microscopy was adopted for observation and image capture. “-” stands for missing.

**Figure 6 viruses-11-00130-f006:**
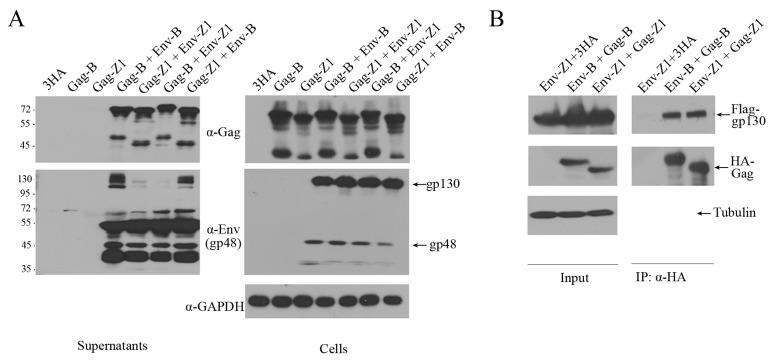
The interaction between Gag and Env. (**A**) HEK293T (4 × 10^6^) cells were transfected with different eukaryotic expression plasmids, 3HA-Gag and 3HA-Env, for 2 days. Cells and supernatants were harvested separately, and the cell culture supernatants were cleared by ultracentrifugation following filtration with a 0.45 μm membrane. Levels of Gag and Env in cells and supernatants were measured by western blotting. (**B**) Immunoprecipitation (with anti-HA) and immunoblot (with anti-HA and anti-Flag) of HEK293T (4 × 10^6^) cells co-transfected with the same source of eukaryotic expression plasmids encoding 3HA-Gag and 3Flag-Env.

**Figure 7 viruses-11-00130-f007:**
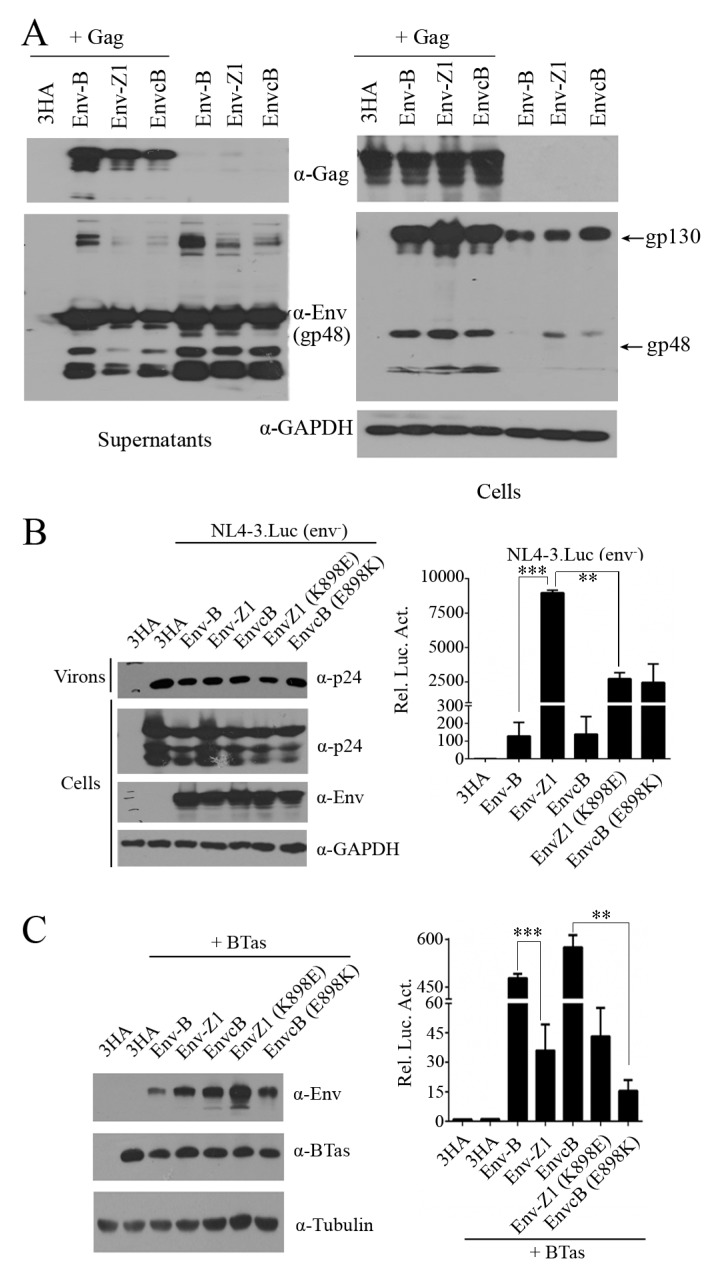
The C-terminal region of Env modulates the entry of cell-free BFV particles but does not affect virus release. (**A**) Immunoblot analysis of Gag and Env in HEK293T cells co-transfected with the indicated plasmids for 48 h and viral particles in supernatants. (**B**) Immunoblot analysis of HEK293T cells transfected with different HA-Env and NL4-3.luc (env^-^). Luciferase reporter assays to analyze the infectivity of viral particles in supernatant, and the data were the averages of three independent experiments. (**C**) Immunoblot analysis of HEK293T cells transfected for 48 h with plasmids encoding different Env and BTas, and luciferase reporter assays to analyze the membrane fusion ability mediated by different Env. The data were the averages of three independent experiments.

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
