# Peer review of "The Influence of Envelope C-Terminus Amino Acid Composition on the Ratio of Cell-Free to Cell-Cell Transmission for Bovine Foamy Virus"

_viruses, 2019, doi:10.3390/v11020130_

Round 1
Reviewer 1 Report
The manuscript entitled “The C-terminus of bovine foamy virus envelope glycoprotein is essential for cell-free transmission” by Zhang et al. describes the C-terminus K898 residue envelope glycoprotein plays important role for entry of the bovine foamy virus which is novel. Overall, the manuscript is presented well but needs minor improvement before accepted in the Viruses.
1. Some of the methods are too short which needs to describe in detail and references should be included where applicable. Some of the methods may include in the supplements if space is limited.
(i) Polyethyleneimine-mediated transfection should describe in detail and/or a reference should be included. Line 75.
(ii) Titration by flourescence microscope is a not good method. A flow cytometry method would yield better quantitation of titers. Line 87.
(iii) Luciferase assay was used for titer quantitation, how was the luciferase activity normalized? Line 99.
(iv) Blocking was done in 50% serum, 6% non-fat dry milk, 15% BSA……………. Is it necessary to include all of the ingredients in blocking buffer?
2. Please describe the advantages of in vitro evolution over other methods in the discussion.
3. In figure 1A, the BFV titer is gradually increasing from p12 to p31 and come to a stationary from p31 to p53. Could you provide an explanation?
4. Figure 1C and 5C are not well visible Please improve the quality of these figures.
5. Compared to supernatants, the co-culture yield multi-fold increase in Luc activity, could you an explanation?
6. The discussion section can be improved by relating the current data with previously published data with proper citations.
Author Response
The manuscript entitled “The C-terminus of bovine foamy virus envelope glycoprotein is essential for cell-free transmission” by Zhang et al. describes the C-terminus K898 residue envelope glycoprotein plays important role for entry of the bovine foamy virus which is novel. Overall, the manuscript is presented well but needs minor improvement before accepted in the Viruses.
1. Some of the methods are too short which needs to describe in detail and references should be included where applicable. Some of the methods may include in the supplements if space is limited.
(i) Polyethylenimine-mediated transfection should describe in detail and/or a reference should be included. Line 75.
A: Thanks for your suggestion. We have revised our statement as advised, please see lines 77-79. The text reads as: “HEK293T, BHK-21, and BICL cells were transfected using polyethylenimines (PEI) (Polysciences, PA) according to the manufacturer’s protocol [33],”.
(ii) Titration by flourescence microscope is a not good method. A flow cytometry method would yield better quantitation of titers. Line 87.
A: Thanks for your suggestion. We have used the same methods to screen the cell-free BFV3026 as reported by others (reference 18: Bao, Q.; Hipp, M.; Hugo, A.; Lei, J.; Liu, Y.; Kehl, T.; Hechler, T.; Lochelt, M., In Vitro Evolution of Bovine Foamy Virus Variants with Enhanced Cell-Free Virus Titers and Transmission. Viruses 2015, 7, (11), 5855-74.). BFV titers were measured by fluorescence microscopy for GFP signals in BICL cells to determine whether it can be passaged, which does not require accurate quantification through flow cytometry method.
(iii) Luciferase assay was used for titer quantitation, how was the luciferase activity normalized? Line 99.
A: We have previously shown that there is a linear relationship between the multiplicity of infection (MOI) of BFV and the activated ratio of luciferase expression in BFVL in our previous article on the BFVL cell line (reference 30: Guo, H. Y.; Liang, Z. B.; Li, Y.; Tan, J.; Chen, Q. M.; Qiao, W. T., A new indicator cell line established to monitor bovine foamy virus infection. Virologica Sinica 2011, 26, (5), 315-23.). Cells in three replicate wells were infected for each infection.
(iv) Blocking was done in 50% serum, 6% non-fat dry milk, 15% BSA……………. Is it necessary to include all of the ingredients in blocking buffer?
A: This blocking buffer has been used for several years in our lab, and it warrants a
very clean background.
2. Please describe the advantages of in vitro evolution over other methods in the
discussion.
A: Thanks for your suggestion. We have revised our description as advised, please see
lines 346-350. The text reads as: “It is known that BFV is highly cell-associated and
spreads mainly through cell-to-cell transmission [17, 18]. Although FFV, SFV and PFV
can transmit by both cell-to-cell and cell-free pathways, the sequence similarity
between BFV and other FVs (FFV, SFV and PFV) is low, and it is thus difficult to
identify the viral genetic determinants through sequence alignment. We have thus
performed in vitro evolution to generate high-titer cell-free BFV variants.”
3. In figure 1A, the BFV titer is gradually increasing from p12 to p31 and come to a
stationary from p31 to p53. Could you provide an explanation?
A: It is possible that the infectivity of this strain reached the plateau on p31 (passage
31) during in vitro evolution.
4. Figure 1C and 5C are not well visible Please improve the quality of these figures.
A: Thanks for your suggestions. We have improved the quality of these figures.
5. Compared to supernatants, the co-culture yield multi-fold increase in Luc activity,
could you an explanation?
A: Although BFV-Z1 clone could produce infectious cell-free BFV, the efficiency of
BFV cell-to-cell transmission was still much higher than cell-free transmission.
6. The discussion section can be improved by relating the current data with previously
published data with proper citations.
A: We have revised the discussion as suggested, please see lines 347, 351, 357 and 361.

Reviewer 2 Report
The first two comments state the major problems with the current manuscript. These MUST be addressed. There are also a number of other points to consider.
1. The data show that Z1 is a better virus for cell-cell and for cell free transmissions than the B clone. This does not mean Z1 is a better virus only for cell free transmission, which is what the authors’ claim and is reflected in the title.
2. A major problem is that when the B clone is transfected there is little to no FV protein in cells, Figure 2. What is wrong with the B clone? This is never mentioned in the text and greatly weakens the paper. The authors must explain in Figure 2, why the B transfection looks so weak compared to Z1. B may simply be a defective virus in all respects. Is the B clone typical of BFV3026? If ten clones are chosen, for example, do they all look like B? Or is B a rare variant?
3. Abstract:Line 9 Each specific FV such as BFV or FFV has a narrow host range in vivo, but in vitro, FV infects all cell types and has a broad host cell range. Thus, the statement regarding FV extensive host range is not clear.
4. Line 10 Use the word “distinguish” not “differ”.
5. Line 19 Change the word “unraveled” to “shown” or “described”.
6. Line 25; 42 Use the word “Orthoretroviruses” rather than “other retroviruses”.
7. Line 39 should start a new paragraph.
8. Authors should more clearly state what they mean by cell-free transmission when first introduced.
9. Line 72 “Expressing” should be used instead of “expression”.
10. Orthoretroviruses can bud without Envelope, however FVs require envelope glycoproteins, and this has been extensively shown. The authors should make this description more succinct in the introduction.
11. Methods:What is the origin of the B virus? This should be clearly stated in the Methods section. Also, BICL cells should be clearly described in the Methods sections.
12. Line 106 p43 is not defined.
13. Line 77 “Titration of…” not “screening...titer determination”.
14. Subviral particles must be clearly defined in the Methods section.
15. Line 108 Centrifugation speed is not given for the Hirt extraction.
16. Line 111 “Floccule” word usage is unclear – change this word.
17. Results: Line 153 As BFV does not induce foci of transformation, titer should be given as infectious units, IU, instead of FFU.
18. As indicated in the Main Point 2, in Figure 2, there is no explanation for the lack of Gag protein in the B clone transfection.
19. The authors should state how the sequences in Z1 and B relate to the BFV in Genbank.
Author Response
The first two comments state the major problems with the current manuscript. These MUST be addressed. There are also a number of other points to consider.
1. The data show that Z1 is a better virus for cell-cell and for cell free transmissions than the B clone. This does not mean Z1 is a better virus only for cell free transmission, which is what the authors’ claim and is reflected in the title.
A: Our data has shown that Z1 is more efficient in cell-cell and for cell free transmissions than the B clone, as this reviewer pointed out, and we have stated this on lines 166-170. The text reads as: “Furthermore, we observed that replication capacity of the Z1 clone was 40 times greater than that of the B clone, as measured by co-culture assay (Figure 1C). Notably, the cell-free Z1 virus particles were infectious (Figure 1C) and spread in a long-term infection (Figure 1D). Together, these data demonstrate that the pBS-BFV-Z1 clone is highly infectious and produces infectious cell-free virus particles.”, The title “The C-terminus of bovine foamy virus envelope glycoprotein is essential for cell-free transmission” only emphasizes the importance of Env protein in cell-free transmission.
2. A major problem is that when the B clone is transfected there is little to no FV protein in cells, Figure 2. What is wrong with the B clone? This is never mentioned in the text and greatly weakens the paper. The authors must explain in Figure 2, why the B transfection looks so weak compared to Z1. B may simply be a defective virus in all respects. Is the B clone typical of BFV3026? If ten clones are chosen, for example, do they all look like B? Or is B a rare variant?
A: The pBS-BFV-B clone was one of the 18 highly active clones, generated by amplifying viral DNA extracted from BFV3026-infected Cf2Th cells (reference 29: Bing, T.; Yu, H.; Li, Y.; Sun, L.; Tan, J.; Geng, Y.; Qiao, W., Characterization of a full-length infectious clone of bovine foamy virus 3026. Virologica Sinica 2014, 29, (2), 94-102.). Our previous results show that the full-length BFV3026 clone B is fully infectious in Cf2Th cells. When the B clone was transfected into BHK-21 for 2 days, we could detect the expression of Gag (the results have added in Figure 1B), but the activity of clone B was weaker than clone Z1 (Figure 1B and 1C), thus when we used same low ratio of infectious cells for virus passaging after 2 days, we could not detect the Gag protein on day 4 in Figure 2B.
3. Abstract: Line 9 Each specific FV such as BFV or FFV has a narrow host range in vivo, but in vitro, FV infects all cell types and has a broad host cell range. Thus, the statement regarding FV extensive host range is not clear.
A: Thanks for your suggestion. We have revised it as advised, and it now reads on lines 9-11. The text reads as: “Foamy viruses (FVs) have extensive cell tropism in vitro, special replication features, and no clinical pathogenicity in naturally or experimentally infected animals, which distinguish them from orthoretroviruses.”.
4. Line 10 Use the word “distinguish” not “differ”.
A: We have revised it as advised, and it now reads on lines 9-11. The text reads as: “Foamy viruses (FVs) have extensive cell tropism in vitro, special replication features, and no clinical pathogenicity in naturally or experimentally infected animals, which distinguish them from orthoretroviruses.”.
5. Line 19 Change the word “unraveled” to “shown” or “described”.
A: We have revised it as advised, and it now reads on lines 19-21. The text reads as: “Taken together, our data have shown the genetic determinants that regulate cell-to-cell and cell-free transmission of BFV.”.
6. Line 25; 42 Use the word “Orthoretroviruses” rather than “other retroviruses”.
A: Thanks for your suggestion. We have revised it as advised, and it now reads on line 11, 26, 43 and 48.
7. Line 39 should start a new paragraph.
A: Thanks for your suggestion. We have revised it as advised.
8. Authors should more clearly state what they mean by cell-free transmission when first introduced.
A: We have revised it as advised, and it now reads on lines 31-32. The text reads as: “Viruses have two major transmission strategies: cell-free transmission, involving the release of virus particles into the extracellular space, and cell-to-cell transmission [9-11].”.
9. Line 72 “Expressing” should be used instead of “expression”.
A: Thanks for your suggestion. We have revised it as advised, and it now reads on lines 75-77. The text reads as: “The plasmids expressing Env and Gag were constructed by inserting the coding sequences of BFV Env and Gag into indicated vectors, including pCMV-3HA and pCE-puro-3Flag.”.
10. Orthoretroviruses can bud without Envelope, however FVs require envelope glycoproteins, and this has been extensively shown. The authors should make this description more succinct in the introduction.
A: Thanks for your suggestion. We have modified the description on lines 45-47. The text reads as: “ FV Gag, lacking the myristoylation membrane targeting signal, cannot produce cell-free Gag-only virus-like particles [18, 24, 25].”.
11. Methods:What is the origin of the B virus? This should be clearly stated in the Methods section. Also, BICL cells should be clearly described in the Methods sections.
A: The pBS-BFV-B clone was one of the 18 highly active clones, generated by amplifying viral DNA extracted from BFV3026-infected Cf2Th cells. We have now described this in the Methods section on lines 70-72. The text reads as: “BFV3026 full-length genomic DNA clone pBS-BFV-B was generated by amplifying viral DNA extracted from BFV3026-infected Cf2Th cells. The BFV infectious clone pBS-BFV-Z1 was constructed using the same methods of pBS-BFV-B as previously reported [29].” We have described the BICL cells in the Methods section on lines 63-64. The text reads as: “and BICL (BHK21-derived indicator cells containing an enhanced green fluorescent protein under the control of the BFV LTR) cells [31, 32]”.
12. Line 106 p43 is not defined.
A: We have revised this as advised, and it now reads on line 109. The text reads as: “BICL cells infected with cell-free BFV p43 (passage 43) were harvested,”.
13. Line 77 “Titration of…” not “screening...titer determination”.
A: Thanks for your suggestion. We have revised this as advised, and it now reads on line 80. The text reads as: “2.3. Titration of cell-free BFV3026”.
14. Subviral particles must be clearly defined in the Methods section.
A: We have revised this as advised, and it now reads on lines 116-119. The text reads as: “2.8. Enrichment of wt and Sub-Viral BFV particles Six milliliters of cell culture supernatants containing BFV particles or SVPs (including Env-only and Gag-Env SVPs) were filtered through 0.45 μm membranes and layered on a 1 mL cushion of 20% sucrose in PBS (w/v).”.
15. Line 108 Centrifugation speed is not given for the Hirt extraction.
A: We have revised this as advised, and it now reads on lines 111-113. The text reads as: “Following centrifugation (10 min at 4,000 rpm), the pellet was dissolved in 400 μL TE and 90 μL 5 M NaCl at 4°C overnight.”.
16. Line 111 “Floccule” word usage is unclear – change this word.
A: Thanks for your suggestion. We have revised this as advised, and it now reads on lines 114-115. The text reads as: “The purified DNA was then washed with 70% ethanol, dissolved in 20 μL TE, and stored at -20°C.”.
17. Results: Line 153 As BFV does not induce foci of transformation, titer should be given as infectious units, IU, instead of FFU.
A: Thanks for your suggestion. We have revised this as advised, and it now reads on lines 156-157. The text reads as: “After 53 passages, the cell-free infectivity of BFV reached a plateau of 105 IU/ml (Figure 1A).”.
18. As indicated in the Main Point 2, in Figure 2, there is no explanation for the lack of Gag protein in the B clone transfection.
A: When the B clone was transfected into BHK-21 for 2 days, we could detect the expression of Gag (the results added in Figure 1B), but the activity of clone B was weaker than clone Z1 (Figure 1B and 1C), thus when we used same low ratio of infectious cells to passage viruses for 2 days, we could not detect the Gag protein on day 4 in Figure 2B. We have added the explanation in lines 196-198. The text reads as: “Notably, the expression of Gag in BFV-B can be detected for 2 days (Figure 1B), but not for 4 days that was passaged one time with low ratio of transfected cells (Figure 2B). One possibility is that the activity of clone B is much weaker than clone Z1 (Figure 1C).”.
19. The authors should state how the sequences in Z1 and B relate to the BFV in Genbank.
A: The two BFV infectious clones (Z1 and B) were derived from BFV3026 (accession number AY134750.1). In order to show the sequence difference among Z1, B and BFV3026, we have now included the aligned sequences of Z1, B and BFV3026 in the supplementary material (Table S1 and Table S2).

Round 2
Reviewer 1 Report
No comments
Reviewer 2 Report
The fact that the Env C-terminus is required for more than cell-free transmission needs to be addressed in the title. The fact that the B clone transfection shows Gag at 2 days and not 4 days needs to be clearly explained.
